

# Extraordinarily corrupt or statistically commonplace? Reproducibility crises may stem from a lack of understanding of outcome probabilities

Caetano Souto-Maior

Basque Center for Applied Mathematics, Bilbao, Spain
Laboratory of Systems Genetics, National Heart Lung and Blood Institute, National Institutes of Health, Bethesda, Maryland, United States

## ABSTRACT

Reports of crises of reproducibility have abounded in the scientific and popular press, and are often attributed to questionable research practices, lack of rigor in protocols, or fraud. On the other hand, it is a known fact that—just like observations in a single biological experiment—outcomes of biological replicates will vary; nevertheless, that variability is rarely assessed formally. Here I argue that some instances of failure to replicate experiments are in fact failures to properly describe the structure of variance. I formalize a hierarchy of distributions that represent the system-level and experiment-level effects, and correctly account for the between-and within-experiment variances, respectively. I also show that this formulation is straightforward to implement and generalize through Bayesian hierarchical models, although it doesn't preclude the use of Frequentist models. One of the main results of this approach is that a set of repetitions of an experiment, instead of being described by irreconcilable string of significant/nonsignificant results, are described and consolidated as a system-level distribution. As a corollary, stronger statements about a system can only be made by analyzing a number of replicates, so I argue that scientists should refrain from making them based on individual experiments.

## BACKGROUND: REPRODUCIBILITY CRISES IN EXPERIMENTAL RESEARCH

In basic biomedical research, it is generally assumed that a scientific finding, once found, is *true* (*e.g.*, something with the infamous $p < 0.05$ significance level). Scientific tradition had good reasons for the assumption: the researcher presenting it has the appropriate expertise, the experiments are well designed and performed, and the methodological description is detailed enough for anyone to reproduce the results. All the checks and titles are in place if the need ever arises to try and reproduce any work, otherwise there would normally be no reason to doubt it. If a natural process exists, it can be observed anytime, by anyone, using the appropriate tools and just enough expertise on the topic (*Henderson, 2020*), and although this is not logical proof of the converse statement ("if its reproducible, it must be

Corresponding author
Caetano Souto-Maior,
caetanosoutomaior@protonmail.com

true"), reproducibility of a scientific result is (or adds to) evidence of the *truth* of a finding (as opposed to an artifact of tools or of the experimenter); it is expected to follow in thoroughly-performed experiments without further questioning. Given that assumption, a number of studies not being replicable is worrisome, and irreproducibility of the majority of the work in a field is a bonafide crisis (*Errington et al., 2021*; *Klein et al., 2018*); in fact it is good reason to question the body of scientific knowledge itself.

In the past decades to centuries many traditions were forced to change in scientific practice (*Beer & Lewis, 1963*), from being an amateur endeavor performed over an undefined amount of time and communicated on printed article to being a professional, public funding-bound activity instantly available worldwide upon publication, to mention only a few changes. Nevertheless, historical contingencies still guide much of modern research, and requiring as well as depositing the same degree of societal trust in "science" today as it was a century ago is not warranted—research is now performed under very different constraints, in addition to having both broader and deeper importance to societal development. In other words, the work performed by the extensive network of research institutions and almost nine million professionals (*Schneegans, Lewis & Straza, 2021*), that forms the basis of a technology-oriented (and possibly obsessed) society, must be stringently verified if we are to justify the expense and reliance on its results.

In that light, reproducibility of results or lack thereof has come to epitomize the very trust in the robustness of science and in scientists themselves, and there are reasons for concern about the quality of current research. Being a profession with merit and career prospects largely based on publications and especially publication-derived metrics, under anything less than ideal conditions it is likely that deleterious incentives will arise. Academia has become increasingly competitive (*Carson, Bartneck & Voges, 2013*), with publication record requirements becoming extremely high and often unrealistic or unreasonable. Arguments have been made for decreasing robustness and novelty of research (*Begley, Buchan & Dirnagl, 2015*; *Chu & Evans, 2021*), the deleterious influence of job structure and lack of stability (*Mirowski, 2012*; *Zollman, 2019*; *Ponzi, 2020*), the increased prevalence of mental health issues among researchers (*Moulin, 2020*; *Woolston, 2019*; *Loissel, 2020*; *Nicholls et al., 2022*; *Eleftheriades, Fiala & Pasic, 2020*) and, more extremely, of a systemic failure of academia that would ultimately be responsible to increasingly irreproducible results (*Lydersen & Langaas, 2021*). While this growing body of evidence about these issues cannot be ignored, and the fact that it likely plays a role in a decreasing trustworthiness of scientific output, I will instead make the argument that there are other reasons for apparently inconsistent results, which unlike the former, are more easily quantifiable and addressable.

None of the arguments above are to say that biomedical science hasn't delivered awesome knowledge and applications over its last century or so of evolution, and there have been counterclaims stating there is no evidence of deterioration in scientific knowledge production. Despite increasingly polarized views between "academia is broken" and "everything is fine"—reality probably is somewhere in between them, and involves recognizing and addressing the inexorable flaws of any human activity (*Leonelli, 2018*;

*Voelkl et al., 2020*; *Karp, 2018*). More to the point, the proposal in this work is essentially orthogonal to that axis of polarization, and the proposal is the following: in most or all experimental research (especially in the life sciences) it is fully expected that any repetition will *not* yield the exact same results, in fact they may yield very different ones even when performed with the same materials, by the same experienced researcher, and only a few weeks or days apart. Differences in repeated measurements can be partly explained by measurement error and uncontrolled or uncontrollable environmental factors, and are statistically incorporated into analyses as implicit test assumptions or explicit variance parameters; therefore, they are formally accounted for within the scope of the experimental design. It is possible to try and control or reduce to some degree this variation, but impossible to eliminate it completely—that is also the reason why sample sizes should be adequate to identify an effect of a certain size, taking in to consideration any previous knowledge of the system and the resources available. This quantitative reality is well-accepted and understood by all (or at least most) experimental scientists in the context of individual measurements, but discrepancies between entire replicate experiments are seen as qualitative—they do not fit that basic statistical framework *as is*, and may be labeled as instances of *irreproducibility*.

I argue here that an apparent qualitative discrepancy can arise from an inadequate interpretation of the structure of variance, and is in fact quantitative and amenable to existing statistical treatments, not unlike that of individual measurements with error. In the next sections I demonstrate how simplistic assumptions can lead to that picture, how to correctly interpret replicate experiments in a general (and intuitive) way, and the implications for integrating information from replicate experiments. This work is based on a full-length article presented at the 9th Conference of the Society for the Philosophy of Science in Practice, in Ghent, Belgium, which was deposited as a preprint for the conference (*Souto-Maior, 2022*).

## NAIVE VIEWS OF EXPERIMENTATION AND REPRODUCIBILITY IN THE NATURAL SCIENCES: "THE *TRUTH* IS OUT THERE"

The most naive, deterministic view of an experiment is that it should always lead to the exact same outcome, the same exact measurement value—*i.e.*, there is a true value $\theta$ that can be observed with infinite or arbitrary precision. That will essentially never happen except for deterministic computer simulations (and even then only up to high-but-finite machine precision), since even the narrowest physical observations will still show some intrinsic variation and high-precision equipment is also always associated to some measurement error. As *Leonelli (2018)* points out, replicability/reproducibility have different meanings as well as usefulness in systems with different expected variation; here I will focus on experimental research, particularly biological systems, where there is clearly noticeable variation between subjects of a sample within any experiment (there may be extreme cases with barely any variation, like a highly-toxic compound at a very high dose, but those are both rare and of little significance).

Under this basic statistical "the-truth-is-out-there" picture, there is still a *true parameter* $\theta$, but it can only be observed with some variance $\sigma^2$ due to sampling—although it is rarely the case that observations are obtained by direct sampling of the parameter (Fig. 1A), any statistical test or model can be formalized to have $\theta$ represent the relevant quantity conceptualized here. Given an experimental design with sample size $N$ it should be possible to identify an *effect* of a certain magnitude at a chosen significance level (*e.g.*, the infamous $\alpha = 0.05$). This general description stems from concepts like the long-run frequency of some kind of observation as well as the related definitions of significance and confidence intervals commonly found in textbooks and associated with *Frequentist Statistics* (*Berger, 2022*), although it is not inexorably linked to the Frequentist practice.

An example of this formulation is a simple linear regression, $y_i = \beta_0 + \beta_1 x_{i1} + \beta_2 x_{i2} + \cdots + \beta_n x_{in} + \varepsilon_i$ where $y_i$ denotes the value of the $i^{th}$ observation and random noise $\varepsilon_i$ is added to each as a draw from a normal distribution. The parameter $\theta = \beta$ can be written as a vector of coefficients $\beta = [\beta_0 \ \beta_1 \ \beta_2 \ \cdots \ \beta_n]$, which yields a compact linear algebra notation with $X$ as the design matrix for the experiment and vector of random noise $\varepsilon$:

$$y = X\beta + \varepsilon$$
$$\varepsilon \sim \mathcal{N}(0, \sigma) \tag{1}$$

In biomedical experimental setups, it is common practice to repeat an experiment and pool the data, but in practice the pooling is only done if the new data "improves" the result (*i.e.*, fits the preferred results of the investigator despite lacking a justifiable criterion, a practice that is questionable at best and fraudulent at worst), which is believed to be by virtue of increased sample size: if the repetitions are qualitatively "similar" (*e.g.*, the effect has same sign and subjectively the magnitude is considered similar) no issue is taken, but if individually they are "different" (*e.g.*, one is significant and the other is not, or they have opposite signs) they are considered inconsistent (Fig. 2A)—in any case, no two results can be formally reconciled this way without some *post hoc* analysis, only subjectively be classified as "agreeing" or "disagreeing".

Here a series of practices that are at least questionable are unfortunately also common: apply different statistical tests and use those where all experiments turn out significant, analyze all combinations of subset pooling and use those that turn out significant, discard supposed outliers (or whole replicates) without justification, and others; I will not deal here with misuse of basic statistical methodology intentionally, by incompetence or neglect of good practice.

Under these premises, the main assumption when analyzing two or more repetitions of the same experiment is that the data generating process is fixed, that is, both the model as well as the parameters are unchanged between experiments because this is the *truth*; if that holds, generating more samples by repetition cannot affect $\theta$ (or $\sigma$), it simply increases with sample size $N$ and with it (for some range $N$, conditional on the particular effect size) the power to obtain significance when testing for an effect. This assumption is convenient and would allow us to invoke the Central Limit Theorem to state that (sample) mean

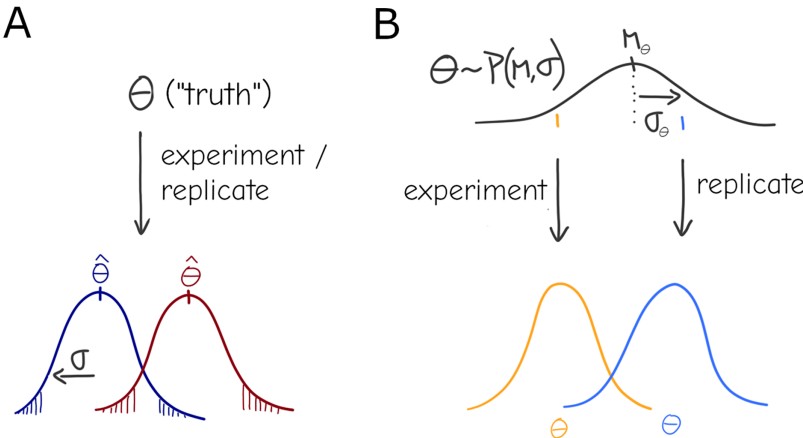

**Figure 1** Under a naive view of experimentation and replication there is a true effect value $\theta$ (*i.e.*, the *parameter*), and different experiments will produce different *point estimates* (Dark red and Dark blue distributions) for the parameter only due to sampling (A). A more general view allows for variation in the parameter: each experiment is a draw from a distribution $P(\theta|\mu, \sigma)$, *i.e.*, the effect of interest is interpreted as a system-level distribution (black) that applies to all experiments—individual results (orange, blue) are expected to vary even under perfect sampling (B).

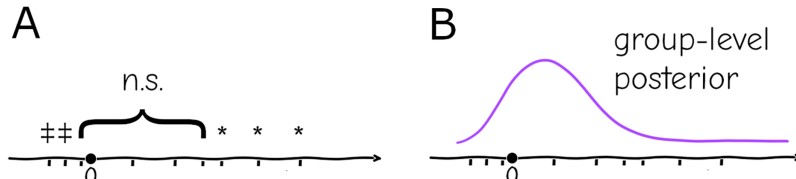

**Figure 2** Replication (ticks along the axis) under a significance testing framework will result in a string of significant (*, or in the opposite direction ‡) and nonsignificant (*n.s.*) outcomes which cannot be quantitatively assessed (A). Under a hierarchical model replication contributes increasingly to the confidence in the group-level distribution, and individual experiments can be assessed against the probabilities given by this distribution (B).

estimated effect can be identified with standard error $s \approx \sigma/\sqrt{N}$, which could be reduced arbitrarily by performing further observations, since the population distribution would be unchanged. Nevertheless, the assumption is usually wrong and, as mentioned above, with no way to reconcile discrepant results the only explanation is irreproducibility by sloppiness, incompetence, fraud (or, at best, bad luck). Adopting a less narrow assumption, however, allows other explanations.

There have been calls, for instance, for the use of formal methods to address replicability by conditioning on the original result and accepting that replication rates vary across fields and types of experiments (*Devezer et al., 2021*); this moves away from the naive view by implicitly abandoning the notion that the same result must be observed each time (although it requires picturing a reality where experiments may be performed an infinite number of times). In the next section I will formalize this broader picture using concepts that are easy to relate to existing statistical methodology.

## THE TRUTH IS UNCERTAIN: INCORPORATING VARIATION INTO BIOLOGICAL EFFECTS

A more general picture relaxes the assumption of a true, fixed $\theta$ for all experiments performed, past or future, instead allowing it to vary between experiments. Some of the formal features of this paradigm are intuitively familiar to any experimenter: individual data points cannot be observed separately at arbitrary times or places and then analyzed together, they must be observed within the same experimental context—measurements produced in different contexts (*e.g.*, by different experimenters, different days of the week, with different equipment, reagents, or protocols) usually are not directly comparable. Similarly, it is universally accepted in the life sciences that treatment and control groups need to be part of the same experiment, performing a measurement on a treatment group and assessing controls at a different occasion will almost certainly produce artifactual and/or invalid results.

Replicates, on the other hand, are by design and necessity performed in different experimental contexts—whether it is repetition by the same experimenter at a later time, or an attempt to reproduce the results by a different laboratory. It is also intuitively known that replicates will not produce the same results, but it is believed that the "broad features" must be conserved between replicates (tough this is usually loosely defined)—in sum: variation is expected because not everything can be controlled, but there must be some uniformity since the same biological mechanisms should be at play (*Glass, 2014*).

The simplest formalization of this would be a mixture, or compound distribution: at the experiment level, deviation $\tau$ is expected around an average $\theta_j$, but that average itself is distributed according to another distribution (Fig. 1B), *i.e.* $\theta \sim P(\mu, \sigma^2)$.

$$\theta \sim P(\mu, \sigma^2)$$
$$x \sim Q(\theta, \tau^2)$$
$$f(x) = \int_{-\infty}^{\infty} q(x|\theta)p(\theta)d\theta$$

where $f$ is the probability density function and the fixed parameters $\tau, \mu, \sigma$ are omitted for clarity.

Staying within a frequentist framework this leads to *random effects* formulations (*Borenstein et al., 2010*). That is, in addition to random error $\varepsilon_i$ for individual observations, further error $\gamma_j$ is associated to each replicate (*i.e.*, the former is different for each observation $i$, the latter is the same for observations within the same replicate $j$, but different between replicates, $\gamma_j \neq \gamma_k \iff j \neq k$). Under the guise of linear regression, for instance, this leads to variable-slopes and/or variable-intercept models; each replicate can have their own slope/intercept because the noise added to the parameters is specific to each repetition; however, the fixed linear coefficients $\beta_l$ themselves are common to all of them. Formally this amounts to including a vector $\gamma \sim \mathcal{N}(0, G)$ in Eq. (1) and a design matrix $Z$ that encode the replicate-specific errors, which for convenience is assumed to be gaussian and do not change $\theta$ on average:

$$y = X\beta + Z\gamma + \varepsilon$$
$$\varepsilon \sim \mathcal{N}(0, \sigma) \tag{2}$$
$$\gamma \sim \mathcal{N}(0, G).$$

As a corollary, seemingly discrepant results are easily explained: having slopes with opposite signs, or different significance can be explained by the random effects formalized as gaussian noise.

While the formulation in 2 is convenient in many ways—with ordinary least squares as a starting point, it can be straightforwardly formalized using compact linear algebra notation to give a closed-form solution—an equivalent and arguably more intuitive description is that of hierarchical Bayesian inference (technically either statistical "philosophy" can be used to formulate the description below, but it is common in Bayesian and rare in Frequentist practice). Under this formalism the "true" experiment-bound parameter $\theta$ that generated the observations can be seen as draws from a distribution of parameters, again $\theta \sim P(\mu, \sigma^2)$—the former parameter can be dubbed individual- or replicate-level effects and the latter are group- or system-level effects that is common to all replicates. In the context of linear models, individual-level effects are equivalent to the varying slopes/intercepts of random effects models. If the distribution of parameters is gaussian, the formulations are equivalent; however, Bayesian practice lends itself to more readily generalizing to an arbitrary choice of distribution, and is arguably more intuitive and visual (Fig. 1).

The hierarchical paradigm readily incorporates the intuitively-understood phenomena mentioned above—*e.g.*, the parameters that generate data are different between replicates due to uncontrolled or uncontrollable variables, but they are not arbitrary or unrelated, they are instances of a common distribution with well-defined mean and finite variance. More importantly, the system-level distribution integrates data from multiple repetitions, and conversely allows specific replicates to be assessed as "typical" or "extreme" according to the probabilities given by the common distribution (Fig. 2B). As mentioned above, a naive fixed-effects approach not only cannot account for this variation between settings, but replication cannot be integrated into a single informative quantity. At a basic level it will cause misinterpretation of repeated experiments within a lab, at a larger scale it will generate false expectations for reproducibility of results.

## A BRIEF CASE STUDY

Analysis of differential expression (DE) of genes relies on measuring counts of RNA transcripts (or possibly other gene products, like protein). These are quantified using technologies like the direct sequencing of RNA molecules (*RNA-seq*), though formerly they used other methods like *microarrays* containing a large number of probes that would anneal to specific sequences (*Lowe et al., 2017*). Linear models are then applied to compare those counts between different experimental groups and assess significance of the differences.

Expression data is known to be noisy and, being a fairly complicated and expensive method, sample sizes are constrained by effort and cost. Entire experiments often consist

of a handful of samples (although each sample produces readouts for thousands of genes, this does little for statistical power). Given the complexity of the systems and characteristics of the methodology it is common to see wide variation in the readouts obtained by different experiments on the same system—once again, it is expected that despite changing values the general *trends* (also and again loosely defined, *e.g.*, expression in one group) is consistently greater than the other. It is also common for that not to be observed. Many analysis packages implement a linear or generalized linear model (GLM) for that purpose, but they normally do not allow for random effects, and therefore require either that replicates are pooled as one, or that they are analyzed separately with independent results obtained for each of them.

In *Souto-Maior, Serrano Negron & Harbison (2023)* a large artificial selection plus RNA-seq experiment was conducted, with three populations: one selected for short sleep, one for long sleep, and one control population (top panel labels), thirteen time points, which was replicated twice, weeks apart. The resulting data set was quite large compared to typical RNA-seq experiments, with a total of 312 samples; furthermore sequencing of samples along generation allows clear visualization (Fig. 3A) and inference of the linear trend along time (as opposed to the difference between treatments a single time point for instance).

Nevertheless, the expanded design does not solve the issue of replication, and in some cases the generation trend is opposite between replicates—the significant negative gene expression slope for short sleepers may turn out to be positive in a new replicate performed correctly. A hierarchical GLM allows each replicate to have their own slope, while the group-level distribution accounts for the range of possible values of the linear coefficients—this is appropriate in this system (under an assumption of additive effects) because selective breeding creates phenotypes by combination and recombination of any genes that may have an effect, and it is not expected that the same genes will be selected to the same extent every time the artificial selection procedure is replicated.

Using this approach on GLMs results in some formulation of generalized linear mixed models (GLMMs), a fairly standard statistical model; nevertheless, *Souto-Maior, Serrano Negron & Harbison (2023)* ultimately find that GLMs are unable to accurately capture the gene expression trends, and use Gaussian processes to model the nonlinear trends in the data. The hierarchical statistical structure is not model-specific nor requires special properties of the model; therefore, it is not restricted to any of class of models. Whether the models are linear, nonlinear, or any whatever its form, this formulation always yields results that allow different replicate-specific parameters constrained by a system-level distribution, and therefore reconciles between-replicate variance.

Replicates have inbuilt sources of variation that are not controlled; therefore, assuming parameters do not change between experiments is a procedure very likely to produce artifactual results. Acknowledging this variation through a system-level distribution of parameters, out of which individual replicates are drawn allows statements about the system to be made, instead of about individual—and potentially contradictory—replicates. Conversely, the system-level distribution may have large variance, in which case statements about its parameters cannot be very precise—increasing the number of

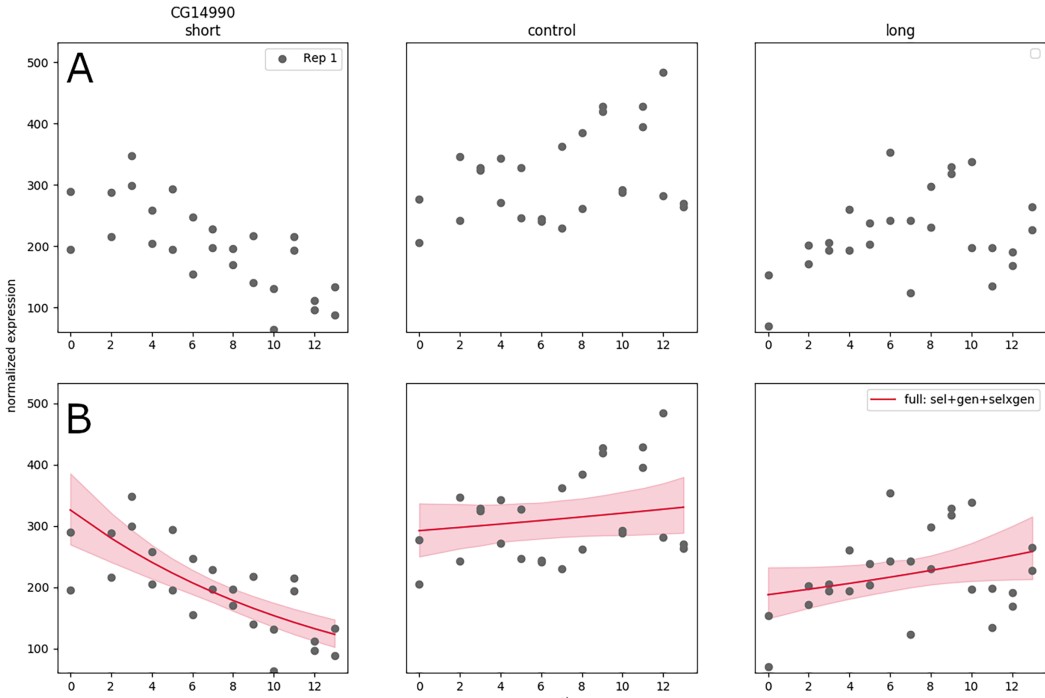

**Figure 3** RNA expression data for a single replicate under different treatments and controls as a function of generation in artificial selection experiment (A). (Generalized) linear model fit to that data showing trends in expression as a function of time (B).

replicates, $R$, will increase this precision, just like increasing number of observations, $N$, will increase within-experiment precision. All data and code necessary to reproduce this case study is available in the original publication of the data and methods (*Souto-Maior, Serrano Negron & Harbison, 2023*).

## IMPLICATIONS AND PERSPECTIVES

Reproducibility is essential to science given the expectation of uniformity in the laws of nature; nevertheless, these same laws of nature are probabilistic at all scales, whether due to properties intrinsic to the system or epistemological constraints. It is not and it cannot be expected that all replication will yield the same result, after all, under the slightest uncertainty "the same" is a subjective label.

There was a time when statistics was not a part of mainstream research, but at this day and age no serious scientist would analyze individual observations separately, without a statistical framework to account for their distribution. Conclusions about an experiment cannot be drawn from a single observation; similarly, conclusions about a system cannot be drawn from a single experiment—in both cases a statement about one level in the hierarchy would be wrongly made if it describes a outcomes at a different level. Like variation between individual observations requires a statistical model, variation between replicates must be formally accounted for and integrated to provide a correct interpretation of system parameters—because the levels are nested, the two levels of

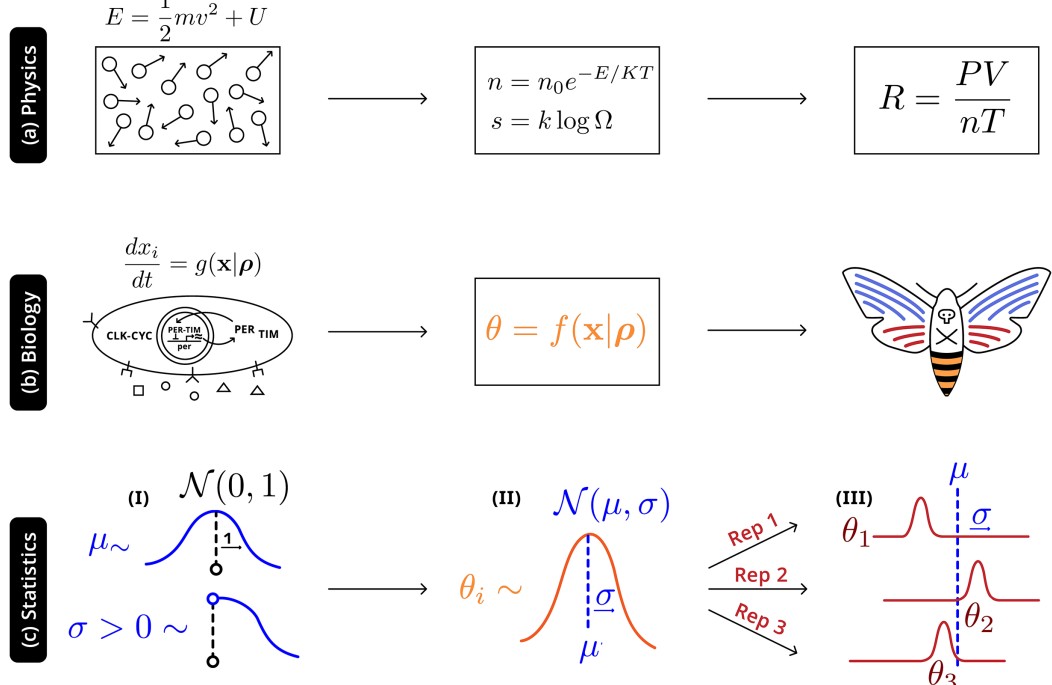

**Figure 4** Robust higher-level phenomena emerge from integrating lower level interactions in (A) physical systems, where laws and universal constants are precisely defined by system state, and (B) biological systems. Statistically, (C) a Bayesian hierarchical structure uses the information of hyperpriors (I) that provide generic information to (II) distributions $\mathcal{N}(0, \sigma)$ of system-level parameters $\theta_i$; this distribution reconciles replicate-specific estimates from disparate experimental settings (III) with deviation $\sigma$ from expectation $\mu$.

estimates conceptualized here (observation and replicate) immediately suggest a hierarchical structure. As a consequence, this formalization marks the transition between a paradigm where replicates agree (or disagree) qualitatively—and interpretation becomes increasingly difficult as data accumulates—to one where all replication contributes to a better understanding of the system, and even opposite results increase our confidence how much variance is expected in the system.

The implications of this formulation go beyond the interpretation of replicates of entire experiments. As long as there are shared processes or parameters between distinct experiments, any observations of the system (provided that the proper metadata is recorded, *e.g.*, time or position) will contribute to a better precision of the system-level distribution. Conversely, arbitrarily large data sets can be put together by replicating whole or parts of experiments or even disjoint observations (for instance time points beyond the original range), all of which can be performed under different settings. While in principle several small, disparate experiments can be integrated to create very large data sets using this framework, large and well-performed experiments will still be valuable, since they will make greater contributions to the system-level distribution (by virtue of having more data points and of having better precision, respectively). On the other hand, it is possible to improve on large experiments by adding smaller experiments to the same analysis, instead of having to repeat a larger experiment in its entirety.

Another application of this framework is in obtaining confidence in estimates where some parameters are conserved but others are expected to be completely independent between settings. That is the case for instance of estimates of the reproductive number ($R_0$) of SARS-CoV-2; this quantity depends both on intrinsic biological parameters (like infectivity of each viral particle) as well as behavioral (contact rates between people), public health policy (vaccination coverage, use of nonpharmaceutical interventions), and possibly geographical (humidity, temperature) variables. Inference of $R_0$ will therefore yield wildly different values if done at different times or places; even if the biological component can be disentangled from the rest, it is more than likely that estimates will still vary between settings. The usual result would then be a series of estimates with little information about which one is "correct" or "better" nor how to compare them. Under a hierarchical framework all data sets could be fit under the same or different models with the common parameter being under a common distribution which would allow estimates to be compared according to their likelihood (Fig. 2) and how much they deviate from the mean.

Illustrated in Fig. 4 is an analogy between this statistical and natural hierarchies: in both cases single observable quantities are not precisely informative of the underlying distribution at more fundamental levels; in both cases it is necessary to adequately account for the connection between the different levels. In physics, macroscopic laws arise from (approximately) deterministic microscopic interactions—there will still be variation all levels, but the most relevant construct is the distribution of energies prescribed by statistical mechanics ($n = n_0 e^{-\frac{E}{kT}}$). In biology randomness must be more thoroughly described: phenotypes are a high-level, noisy manifestations arising from several underlying molecular processes (well-defined causal relationships and parameters), with additional variation due to uncontrolled environmental factors, which are then observed through individual experiments. The statistical hierarchy is the simplest in that it only requires the acknowledgement of variation at a level above that of the individual experiment—accurately making inferences about the system will also require describing the natural hierarchies, but that modeling exercise and the interactions with the statistical framework is beyond the scope of this work.

The most important message is this: it is not only inaccurate but wrong to make statements about a higher level (and not directly accessible) in the hierarchy from single (and randomly biased) observations at a lower level. It is also not enough to make repeated observations without an adequate consideration of the structure of variance. The relationship between the system itself and individual observation instances is hierarchical and must be accounted for; issues of repeatability and replicability crises can only be identified if they correctly describe the relationship between the replicates.

## ACKNOWLEDGEMENTS

I would like to thank Susan Harbison and María-Xosé Rodríguez Álvarez for their support and feedback. I would also like to acknowledge members of the Applied Statistics research line and trainees in other groups at the Basque Center for Applied Mathematics for the feedback on internal talks, and attendees at the Society for Philosophy of Science in

Practice 9<sup>th</sup> Biennial Meeting (SPSP 2022) for comments and questions on the presentation and draft presented as part of the conference.

### Funding
This work was funded by the Intramural Research Program of the National Institutes of Health, the National Heart Lung and Blood Institute and by the Ministerio de Ciencia y Innovación (MICINN, AEI) of the Spanish Government through BCAM Severo Ochoa accreditation CEX2021-001142-S and PID2022-136585NB-C22 grant. This work was also funded by the BERC 2022-2025 Program and the grant BMTF (Mathematical Modelling Applied to Health Project)—all funded by the Basque Government. There was no additional external funding received for this study. The funders had no role in study design, data collection and analysis, decision to publish, or preparation of the manuscript.

### Grant Disclosures
The following grant information was disclosed by the authors:
Intramural Research Program of the National Institutes of Health.
National Heart Lung and Blood Institute.
Ministerio de Ciencia y Innovación (MICINN, AEI).
Spanish Government BCAM Severo Ochoa accreditation: CEX2021-001142-S and PID2022-136585NB-C22.
BERC 2022–2025 Program.
BMTF grant.
Basque Government.

### Competing Interests
The author declares that they have no competing interests.

### Author Contributions
- Caetano Souto-Maior analyzed the data, prepared figures and/or tables, authored or reviewed drafts of the article, and approved the final draft.

### Data Availability
   The raw data is available at GitHub: https://github.com/caesoma/Multiple-shifts-in-gene-network-interactions-shape-phenotypes-of-Drosophila-melanogaster.
   The data was originally available from NCBI GEO: GSE202600.

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
