# Peer review of "Extraordinarily corrupt or statistically commonplace? Reproducibility crises may stem from a lack of understanding of outcome probabilities"

_PeerJ, doi:10.7717/peerj.18972_

## Round 0.1 · original submission · Major Revisions

Both reviewers call for greater clarity in the MS. The data from the Case Study especially needs to be linked to and explained in more depth. Reviewer 2 feels there are good points, but Reviewer 1 is not sure you have pinpointed a gap in the existing literature - if so you need to make this clearer.

Reviewer 1 ·

Basic reporting

As a general comment, while I am an experienced biostatistician, I found myself struggling to always link together the arguments that the author is clearly enthusiastic about showing to the reader. There are some very useful points made along the way, but these seemed to me to be, to some extent at least, known already, and at the end of re-reading the manuscript, I still wasn’t sure what I would look at doing differently going forward or how I could use the ideas presented here to further develop methods/approaches. I’ll make more specific comments here and then come back to some general ones later in the review.

The conclusion in the abstract (“stronger statements about a system can only be made by analyzing a number of replicates, so I argue that scientists should refrain from making them based on individual experiments.”, Lines 23–25) advocates for a state of affairs that seems to me to be well reflected in the treatment of systematic reviews and meta-analyses as a higher level of evidence, and umbrella reviews as higher still. I’m not suggesting references in the abstract, but I think that the awareness of evidence synthesis is, at least in fields where I have familiarity, relatively well-accepted already.

Perhaps relatedly, the opening sentence (“It is assumed almost universally that a scientific finding, once found, must be true.”, Line 28) is making a very strong claim, and one that I think that needs, preferably multiple, citations. Again, this doesn’t match my experiences in particular fields, and I think that most fields will have their own examples of well-established knowledge being overturned by new studies.

It is unclear to me which statements in the Introduction are supported by which citations and which are the author’s own opinions (with an exception of the fourth paragraph that does not include any citations at all; the lack of citations in the fifth and final paragraph in the introduction is of course fine as this clearly signals the author’s views). If the author wishes to say “sometimes compared to a ponzi scheme”, for example, readers should be able to see exactly who is said to have made this comparison rather than the more than four line sentence (which ranges from power imbalances through to suicide rates) ending with a collection of four citations.

I might have anticipated more recent citations from philosophers of science who have much to say about the state of research. I also might have anticipated more discussion of the other side, the benefits that society has received from research as it has been and is conducted. This is not a defense of the status quo, well-justified criticisms of which abound, but rather noting that an alternative ought to accompany such criticisms.

One option here might be to focus the introduction on the question of replication itself as around Lines 67–69, the author appears to set some of the introduction aside to focus on their interests.

It seems odd, given the topic, that Ioannidis’ impactful work does not appear to be mentioned.

In passing, I have to strongly disagree with the statement on Line 80 that “sample sizes should be as large as possible”, which would be a position that I feel it would be difficult to justify from an ethical perspective as well as from a resource allocation one.

There are some apparent misunderstandings, such as the CLT (Lines 132–133) and power (e.g., Line 193) that I feel would need attention, although these do not appear to me to impact on the rest of the manuscript and so might not be needed. Similar to these examples is the surprising (to me) claim that most analysis packages do not allow for random effects (Lines 197–200). Could the author be clear about what functionality they see as missing from some specific commonly used statistical packages?

The figures would need to be reworked into more polished versions.

Experimental design

The goal of better understanding replication is always worthy of additional attention, although there has been much work in this area (including that by and following Ioannidis) that is not presented here. Some of the goals here would seem to be addressed in part by meta-regression, which is part of the standard (bio)statistician toolkit when synthesizing results.

I wasn’t entirely sure what the author wanted the reader to take from the (described as ‘brief’) case study and wondered if the author needed to expand this section.

Perhaps a simulation-based approach would be helpful in explaining how they propose their approach to be used and what properties it would have (compared to some standard practices) before the specific (and I would suggest, more detailed) case study is presented?

There are some interesting points that are touched upon, such as what constitutes ‘replication’ that it would be interesting to follow-up/expand on.

Validity of the findings

As a biostatistician, I have to admit that I struggled to see exactly what gap the author is proposing that their ideas would address, nor could I see how these ideas could be implemented going forward. I wondered how their ideas would fit in alongside Bayesian meta-regression, for example.

Additional comments

While there were some useful points made, and I can see potential for expanding on those points, the manuscript seemed to me to lack a clear focus around the gap in the literature that the work would address and the implications of the work for future research and scientific practice.

·

Basic reporting

This is an important article that advocates for changes to statistical practice that focus on quantifying variance between and within experiments in relation to system- and experiment-level effects for better assessing the quality and success of replication studies. This article satisfies most aspects of basic reporting. The language is clear and professional, literature is properly cited to give adequate background, however Figures 1, 2, and 4 appear to be hand-drawn. This isn't necessarily an issue, as they are legible and make good sense, though I wouldn't call them "professional." The article also includes a secondary analysis of data from Souto-Maior et al., 2023, and the data does not appear to be shared in the current article and there is no provided link to access it. I also did not see references to shared code used to analyze this data. The results are self-contained and relevant to the hypotheses, but would benefit from deeper explanation and references to Figure 3 to support the main arguments. There is also some relevant literature the author may find helpful to cite:

https://www.cambridge.org/core/journals/behavioral-and-brain-sciences/article/abs/generalizability-crisis/AD386115BA539A759ACB3093760F4824

https://evidence.nejm.org/doi/abs/10.1056/EVIDoa2300003

Experimental design

This paper is within the aims and scope of PeerJ as a methods article. The article doesn’t have a research question as much as an approach/perspective it wants to promote, using a case study to test and validate it. This approach/perspective is that scientists should use hierarchical modeling approaches to account for variance between and within experiments, quantify system-level and experiment-level effects, and this can help reconcile replication failures.

This perspective is important and well-supported by the arguments and the case study presented in the paper. The methods in the paper are well-described and explained. However, the following changes would greatly help the paper: 1) Share code and data for the case study presented, 2) Include at least one more case study, with data and code, 3) Include simulations illustrating the principles you’re highlighting about why accounting for between- and within-experiment variance for system- and experiment-level effects. I think the ideas in the article are great, but it would be good if they were presented with more empirical support and case studies, along with data/code to replicate these case studies. This code could also be repurposed as starter code that scientists could use to implement the changes that the author advocates.

Validity of the findings

The findings from the case study appear valid, though the data come from a previously published study and are used in a secondary analysis. The author should provide a link or other way to access the data, as well as code to run the analyses presented in the paper. Ideally, the author will include additional case studies (at least one), as well as simulations with code to illustrate the ideas and perspectives they are advocating, so that scientists can best understand the importance and actionability of the author’s insights and repurpose the provided code for their own studies. The conclusions are appropriately stated, and are aligned with the results of the study. The data from the case study is shown in Figure 3, and the only reference to the figure in the article is about visualizing a linear trend in the data over time. It would be great if there were additional references to the figure to support the arguments of the author. Please provide clear explanations of the figure, and how they support the arguments the author is making about needed changes to statistical practice. Also, please explain what “short”, “control”, and “long” mean in the figure.

Additional comments

Overall, this is an important article that advocates for changes to statistical practice that I think would benefit the community. However, the case study is not explained in adequate depth, data or links to the data are not provided, and the same goes for code. The article would benefit from deeper reporting of the results, additional case studies and simulations supporting the main ideas, and sharing data and code.

I have some additional comments related to typographical changes and other pieces of text:
- Abstract: “One of the main result” - “result” should be “results”
- Section 1: “It is assumed almost universally that a scientific finding, once found, must be true.” This is a strong statement without a reference. If you can’t support it with a reference, please remove it.
- Section 1: “must be stringently verified if we are to justify the expense and and reliance of its results.” Please remove one of the instances of “and” and change “reliance of” to “reliance on.”
- Section 1: “It is possible to try and control or reduce to some degree this variation, but impossible to eliminate it completely – that is also the reason why sample sizes should be as large as possible.” Sample sizes should give adequate power for specific Type I and II error rates and effect sizes of interest, but not be as large as possible, as some experiments can be overpowered and wasteful of resources at certain sample sizes. Please rewrite this sentence.
- Figure 1 - The legend includes a clear reference to the distributions shown in panel B (i.e., “(orange, blue)” but not for panel A. Please add this for clarity.
- Section 2 - “in practice the pooling is only done if the new data ”improves” the result,” I believe the author meant “yielding a statistically significant and publishable result” by “improves” but please make this explicit for readers.
- Section 2 - “apply different statistical tests and use those where all experiments turn out significant, analyze” You may want to mention the term “p-hacking” here as a helpful keyword readers can learn and remember.
- Section 2, same paragraph - “discard supposed outliers (or whole replicates) without justification...” Very glad you mentioned this.
- Section 2 - “Under this premises,” Either change “this” to “these” or remove the “s” at the end of “premises.”
- Section 2 - “simply increases N and with it the power obtain significance when testing for an effect.” Insert “to” between “power” and “obtain.”
- Section 2 - “this moves away form the naive view by implicitly abandoning the notion that the same result must be observed each time” I believe “form” should be “from”.
- Section 3, title - “THE TRUTH IN UNCERTAIN: INCORPORATING VARIATION INTO BIOLOGICAL EFFECTS” - Not sure if “Uncertain” should be “Uncertainty” or if “In” should be “Is.”
- Section 3 - “individual data points cannot be observed separately at arbitrary times or places an then analyzed together” I believe “an” should be “and”
- Section 4 - “see for instance Figure 2 of that reference” The reference is given in the preceding paragraph, but please replace “that reference” with the same reference.
- Section 5 - “Finally, illustrates in Fig. 4 an analogy between this statistical and natural hierarchies:” Maybe change this to “Figure 4 illustrates an analogy...” or something like that.

---

## Round 0.2 · Major Revisions

Dear Dr. Souto-Major,

Although reviewer 2 finds your MS improved, reviewer 1 still calls for major revisions. Please address their concerns.

Reviewer 1 ·

Basic reporting

I appreciate the changes that have been made, although these are not as extensive as I would have anticipated, especially in what I see as the most important section, namely Section 5 on Implications and Perspectives. I also appreciated the author’s mainly constructive responses. At the same time, I’m still unclear how readers should respond to the work.

I now better understand the author’s intentions in terms of their target audience. I suggest that this is always clearly signposted. E.g., Lines 15–16 “outcomes of replicates will vary; nevertheless, that variability is rarely assessed” should perhaps be followed by “in biological research” (or something similar). Nothing preceded this that would provide this context for me. I appreciated the qualification in the first sentence of the body of the manuscript (Line 28) and similar elsewhere.

I think that most of the paragraph covered by Lines 50–70 should be considered for removal. This text, using what I would describe as provocative rather than professional language, doesn’t appear to establish anything necessary to the thesis of the work as a whole and I think it is likely to be distracting to readers. The first sentence would seem entirely adequate to making the point that sometimes the variability that leads to mistrust is exactly what should be expected. The discussions here of researchers driven to suicide and the abuses by supervisors, etc., in my view, does not belong here. A similar point could be applied to Lines 71–mid 77. If this were a manuscript advocating for better treatment of, particularly junior, researchers, I would agree with the points, although I’d still suggest a different tone for the language. Here, these points, however, do not appear to support or add context to the point that results will vary from the original experiment to replicate to replicate; if anything they suggest to me that researchers could be driven to unnatural degrees of variability in findings by, for example, fraud in either trying to conform to existing findings or in trying to create novel results.

While in a lecture, I would find the figures appealing, I still don’t see them as ‘professional’ and recommend the author investigate alternative ways of creating them to achieve this.

Experimental design

As Reviewer #2 stated, an experiment is designed according to acceptable type I and type II (conditional on an effect size of interest) errors. Having the opportunity to make a study even bigger that needed for these because the larger sample size was feasible (Line 85) would still constitute ‘research waste’ in my view.

I think that the simulations I discussed previously would help readers, including myself, understand the message. The reproducibility of findings has been well researched elsewhere and providing simulations framed within biological research could only help to persuade the reader and allow them to calibrate their intuitions. I don’t think that this part of the story should be left as an exercise for the reader. At the moment, the manuscript presents an explanations of regression models (Lines 117–121, Lines 169) which the reader doesn’t seem to need to understand what follows them. These could perhaps be the starting point for simulations, leading to the sampling distribution, then to perturbations of this by other experimental factors, and thus to the degree of variability in point estimates from experiments ostensibly estimating the same thing.

The point about power on Lines 139–140 would need to make it clear that this power is conditional on a particular effect. The following point on the CLT isn’t correct. In particular, the CLT does not follow from the assumption that seems to be being intended here, namely that there is a fixed effect size.

Lines 187–189 require references.

Validity of the findings

As the manuscript stands, I’m still finding myself wondering what impact it would have on my work, which is in both clinical and biomedical sciences and so includes work of the type the author is speaking to.

·

Basic reporting

The reporting in the paper is clear, and reviewer comments have been answered adequately. Professional English is used throughout the paper. References, background and context are adequate. Professional article structure is used, and no raw data have been generated for this paper. However, the author should include links in the article to the phenotypic sleep data, code, and omics data used in their previous work which is shown as a case study in the current article. There isn't a hypothesis tested here as much as an approach advocated for, but regardless the ideas and results are self-contained.

Experimental design

The research question is relevant and meaningful and the purpose of this article is clear. The arguments, ideas, and results are rigorous and to high technical and ethical standards.

Validity of the findings

As stated above, all data and code used in the case study should be linked to in the current article to make it easier for readers to follow up and apply the approaches advocated for in the current article. Alternatively, the author could write somewhere in the article that all data and code are available and then cite the reference. Please just make it clear that all data and code from the case study are accessible from the original article.

Additional comments

A few remaining points about language and typos:
-"In basic biomedical research, it is generally assumed that a scientific finding, once found, is true." - I think you can more clearly define what "found" would mean. Do you mean p<0.05? Something else? I think this should be less ambiguous.

-"It is possible to try and control or reduce to some degree this variation, but impossible to eliminate it completely – that is also the reason why sample sizes should be as large as feasible." - The change from "possible" to "feasible" is good but is still a bit loose and ambiguous. Given the author is a Bayesian, they can appeal to sample size justification methods such as estimating precise credible intervals, maximizing utility or something decision theoretical, or other non-frequentist approaches to sample size justification (for the frequentist approach, see, https://doi.org/10.1525/collabra.33267). "Feasible" is not too different from "possible" and could be clearer and more technical.

-"a practice that is questionable and best and fraudulent at worst" - I think "and best" is supposed to be "at best"

-"since the population distribution would be inchanged" - I think "inchanged" is supposed to be "unchanged"

-"and one control population (top panel labvels)" - I think "labvels" is supposed to be "labels"

---

## Round 0.3 · accepted · Accept

Comments made by the second reviewer have been addressed but as you know, Reviewer 1 was not satisfied. As the Editor, I find the tone of the article has improved. As I believe PeerJ should support open debate on issues, so I have recommended Acceptance, but have asked that the Section Editors check my judgement.